# Asymmetric inheritance of centrosomes maintains stem cell properties in human neural progenitor cells

Lars N Royall[1], Diana Machado[1], Sebastian Jessberger[1,2]*, Annina Denoth-Lippuner[1]*

[1]Laboratory of Neural Plasticity, Faculties of Medicine and Science, Brain Research Institute, University of Zurich, Zurich, Switzerland; [2]University Research Priority Program (URPP), Adaptive Brain Circuits in Development and Learning (AdaBD), University of Zurich, Zurich, Switzerland

*For correspondence:
jessberger@hifo.uzh.ch (SJ);
denoth@hifo.uzh.ch (AD-L)

**Competing interest:** The authors declare that no competing interests exist.

**Abstract** During human forebrain development, neural progenitor cells (NPCs) in the ventricular zone (VZ) undergo asymmetric cell divisions to produce a self-renewed progenitor cell, maintaining the potential to go through additional rounds of cell divisions, and differentiating daughter cells, populating the developing cortex. Previous work in the embryonic rodent brain suggested that the preferential inheritance of the pre-existing (older) centrosome to the self-renewed progenitor cell is required to maintain stem cell properties, ensuring proper neurogenesis. If asymmetric segregation of centrosomes occurs in NPCs of the developing human brain, which depends on unique molecular regulators and species-specific cellular composition, remains unknown. Using a novel, recombination-induced tag exchange-based genetic tool to birthdate and track the segregation of centrosomes over multiple cell divisions in human embryonic stem cell-derived regionalised forebrain organoids, we show the preferential inheritance of the older mother centrosome towards self-renewed NPCs. Aberration of asymmetric segregation of centrosomes by genetic manipulation of the centrosomal, microtubule-associated protein Ninein alters fate decisions of NPCs and their maintenance in the VZ of human cortical organoids. Thus, the data described here use a novel genetic approach to birthdate centrosomes in human cells and identify asymmetric inheritance of centrosomes as a mechanism to maintain self-renewal properties and to ensure proper neurogenesis in human NPCs.

## Editor's evaluation

The fundamental work that shows the preferential inheritance of the older centrosomes by the self-renewing daughter cells in human is supported by strong evidence. The findings will be of interest to developmental neurobiologists, but also more broadly to cell and developmental biologists.

## Introduction

During human brain development, neural progenitor cells (NPCs) undergo two modes of cell division. At first, NPCs, at this developmental stage called neuroepithelial cells, undergo expansive symmetric divisions (*Cadwell et al., 2019*; *Libé-Philippot and Vanderhaeghen, 2021*). Symmetric divisions are characterised by the generation of two daughter cells of similar fate; here, two NPCs both retain their potency, their capacity to self-renew and remain in the ventricular zone (VZ). Around gestational week (GW) 5 in the developing human brain (corresponding approximately to embryonic day (E) 11.5 in the mouse embryo), NPCs, at this developmental stage referred to as radial glia or apical progenitor

cells, transition from an expansive phase into a neurogenic phase and shift their mode of division from symmetric to asymmetric divisions (*Götz and Barde, 2005*; *Malatesta et al., 2008*; *De Juan Romero and Borrell, 2015*). Asymmetric divisions result in two daughter cells with different fates and cellular behaviour: one daughter remains in the VZ and retains the ability to self-renew, comparable to the mother cell. The other daughter cell migrates along the basal process of their sister cell, out of the VZ and either directly begins the process of neuronal differentiation or initiates additional rounds of symmetric, differentiating cell divisions producing two neurons (at this stage referred to as intermediate progenitors or basal progenitor cells) (*Noctor et al., 2004*; *Hansen et al., 2010*; *Gao et al., 2014*). Exactly how differential fates of sister cells are established remains not completely elucidated; however, previous work showed that the centrosome might play a central role in distinct fates of daughter cells upon neurogenic, asymmetric cell divisions (*Wang et al., 2009*).

Centrosomes are unbound organelles comprised of two centrioles connected by a flexible linker and surrounded by a dynamic protein matrix called pericentriolar material. Centrosomes are the primary microtubule organising centres of metazoan cells and provide the contractile forces required for mitosis in most human cells (*Bornens, 2002*). The centrosome duplicates once per cell cycle that occurs in a semi-conservative way, producing one centrosome that is older and functionally more mature than the other one. At mitosis, the older and younger centrosome will always be asymmetrically segregated to one of the two daughter cells. Previous work showed that there is non-random inheritance of centrosomes based on their age in the developing cortex of flies (*Januschke et al., 2011*; *Ranjan et al., 2019*; *Sunchu and Cabernard, 2020*), chickens (*Tozer et al., 2017*), and mice (*Wang et al., 2009*; *Paridaen et al., 2013*). Indeed, in the mouse developing cortex self-renewing NPCs appear to inherit the older, more mature centrosome while the newborn neuron inherits the new, daughter centrosome (*Wang et al., 2009*). Notably, randomisation of centrosome inheritance leads to premature depletion of NPCs from the VZ, indicating functional relevance of asymmetric centrosome inheritance during mouse cortical development (*Wang et al., 2009*). How asymmetric centrosome inheritance affects cellular fate is only poorly understood but, for example, recent work showed that Mind-bomb1, a Notch signalling regulator, is preferentially enriched at the younger centrosome, asymmetrically segregates with it into the differentiating daughter cell, and thereby potentially promotes stemness via Notch signalling activation in surrounding NPCs (*Tozer et al., 2017*).

Despite substantial evidence supporting the importance of asymmetric centrosome inheritance in NPCs from diverse, evolutionary distant species, there is little known about human tissues. This is mainly due to the difficulty obtaining samples of human developing cortex coupled with the fact that perturbations are only feasible in vitro. However, it is clear that the human brain relies on species-specific molecular regulators and genetic pathways (*Bystron et al., 2008*; *Pinson and Huttner, 2021*). Recent methodological advances such as the generation of brain organoids, derived from human pluripotent stem cells, recapitulate early steps of human brain development, allowing for novel approaches to characterise the principles of cortical development in human tissues (*Lancaster et al., 2013*; *Di Lullo and Kriegstein, 2017*). Cortical organoids render a perfect system to study asymmetric cell divisions as the stem cells and their progeny segregate spatially with the NPCs lining the ventricle and the neurons surrounding the NPCs (*Qian et al., 2016*). Here, we used human embryonic stem cell (hESC)-derived, regionalised forebrain organoids as a model of early human cortical development to identify the dynamics and functional relevance of asymmetric centrosome inheritance for human neurogenesis.

## Results

### Centriolin-RITE birthdates human centrosomes

With the aim to birthdate and track the segregation of centrosomes over multiple cell divisions in human cells we created a novel genetic tool based on the recombination-induced tag exchange (RITE) system (*Hotz et al., 2012*), consisting of the red fluorescent protein tdTomato flanked by LoxP sites and followed by a second section containing the green fluorescent protein NeonGreen (with each section ending with a terminal stop codon followed by a 3'UTR or a T2A Neo, respectively) (*Figure 1A*). Using CRISPR/Cas9 (*Ran et al., 2013*) we inserted the RITE construct into hESCs immediately upstream of the terminal stop codon of Centriolin, a protein that localises to the subdistal appendages of the mother centriole (*Gromley et al., 2003*; *Gromley et al., 2005*; *Kashihara et al.,*

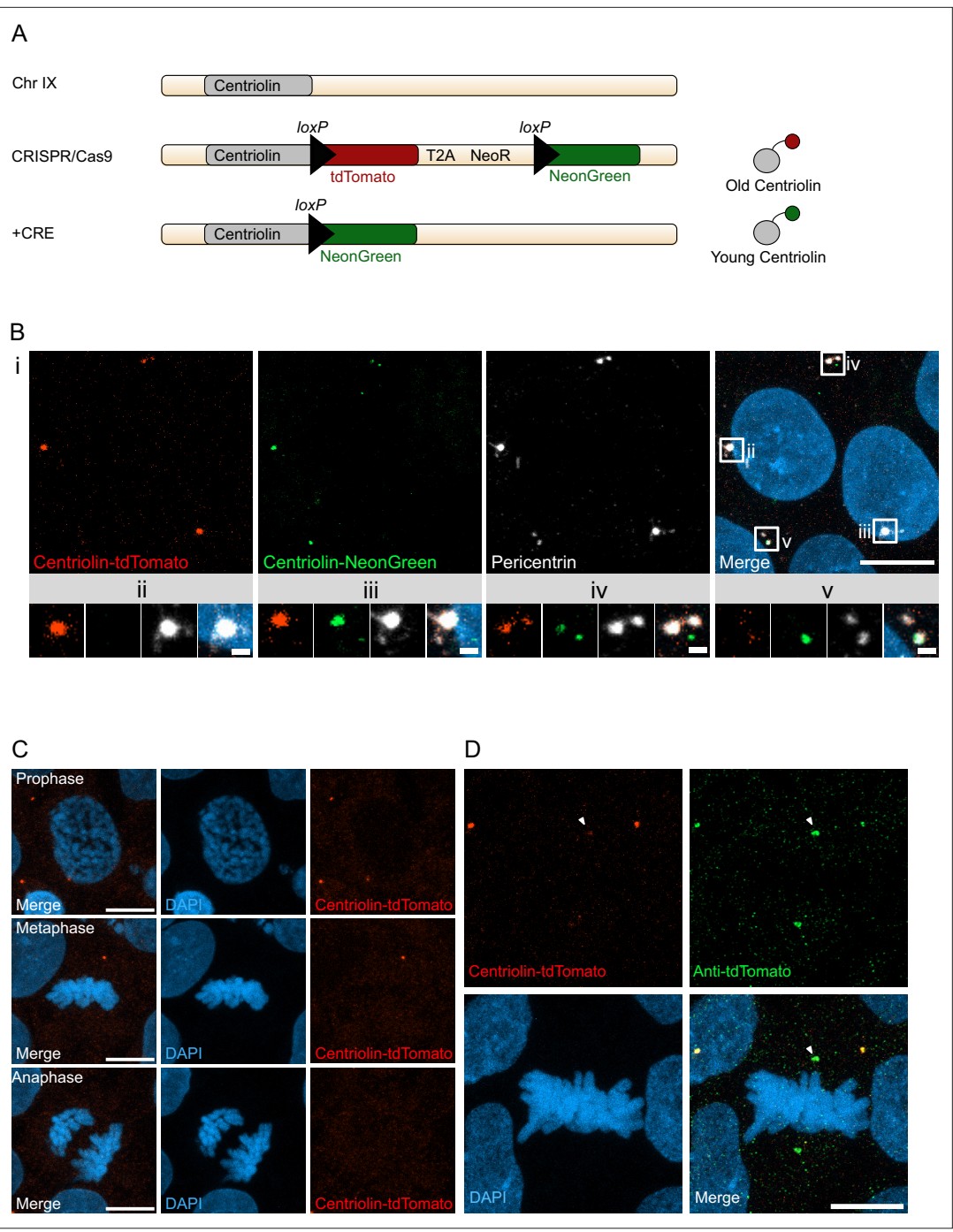

**Figure 1.** Recombination-induced tag exchange (RITE)-based birthdating of centrosomes. (**A**) Centriolin is tagged with the RITE system in human embryonic stem cells (hESCs) so that all Centriolin protein made is tdTomato-tagged. Upon recombination with Cre recombinase, the tdTomato fluorophore is replaced with NeonGreen, making all new Centriolin protein tagged with NeonGreen. Identifying a proteins tag will indicate the proteins age with tdTomato being the oldest, then NeonGreen. (**B**) Image of hESCs expressing Centriolin-RITE 24 hr post Cre induction. Magnified images show a centrosome that has not recombined (ii), has recombined and contains red and green Centriolin (iii and iv), or only green Centriolin (v). Scale bars, 10 µm (upper panel), 1 µm (magnifications). (**C**) Signal of Centriolin through mitosis. In prophase centriolin signal is present, however the signal disappears in metaphase and anaphase. Scale bars, 10 µm. (**D**) Staining for tdTomato protein shows that Centriolin-tdTomato is present at the centrosome during metaphase but that the signal is subsequently quenched. Scale bars, 10 µm.

The online version of this article includes the following figure supplement(s) for figure 1:

**Figure supplement 1.** Centrosome analyses in human embryonic stem cells (hESCs).

*2019*; *Chong et al., 2020*). The Centriolin yeast ortholog, Nud1, has previously been shown to have limited turnover on the pre-existing yeast spindle pole body (*Lengefeld et al., 2017*). As the reading frame is maintained, Centriolin will be constitutively tagged with tdTomato. LoxP recombination with Cre recombinase will excise the first section containing tdTomato, causing Centriolin to be tagged with NeonGreen, and allowing for the discrimination of pre-existing vs. newly synthesised Centriolin based on red vs. green fluorescence. Indeed, tdTomato-positive dots were observed in each cell and NeonGreen was detectable within 24 hr after recombination induced by electroporation of Cre recombinase expressing plasmid; staining with the centrosomal marker protein Pericentrin showed that Centriolin-tdTomato was properly localising to the centrosome (*Figure 1B*). Correct localisation of RITE-tagged Centriolin was confirmed by live imaging of hESCs following electroporation of GFP-Centrin-1, which localised to the centrioles (*Figure 1—figure supplement 1A*). In line with the known behaviour of Centriolin as a subdistal appendage of the mother centriole, Centriolin-tdTomato co-localised to one of the two centrioles.

Examination of cells at different stages of mitosis showed that Centriolin-tdTomato signal was present in prophase cells but diminished by metaphase (*Figure 1C*). To investigate whether this was due to loss of Centriolin-tdTomato protein or quenching of fluorescence, we stained for tdTomato, which revealed that Centriolin-tdTomato protein remained localised to the centrosome throughout mitosis (*Figure 1D*), indicating that RITE-tagged Centriolin allows for tracking centrosomes in human cells. To simplify and facilitate Cre recombinase-mediated recombination without the need of transfection or electroporation we used CRISPR/Cas9 to introduce a stable expression cassette of $ER^{T2}$-CRE-$ER^{T2}$ from the human *safe harbour* locus Adeno-associated virus site 1 (AAVS1; *Roemer et al., 2016*), which we used for subsequent experiments.

## Centriolin-RITE localisation and recombination in human forebrain organoids

We used a heterozygous Centriolin-RITE, $ER^{T2}$-CRE-$ER^{T2}$-positive hESC line to generate regionalised, forebrain organoids (*Figure 2—figure supplement 1A*) to understand how centriolin localises within the three-dimensional structure of human neural tissues (*Qian et al., 2016*; *Denoth-Lippuner et al., 2021*). Day 35 organoids were fixed and stained with the centrosomal marker CEP164 to check for correct co-localisation of RITE-tagged Centriolin within organoids. Indeed, CEP164 co-staining confirmed co-localisation with Centriolin-tdTomato (*Figure 2A*). Ventricle-like structures at the centre of cortical units were easily identifiable by the clustering of Centriolin-tdTomato-labelled centrosomes, belonging to SOX2-positive, NPCs in the VZ (*Figure 2A*). Centrosomes outside of the VZ were more sparsely distributed.

To test whether Centriolin-RITE organoids maintain their capacity to recombine, day 35 organoids were incubated with 4-OH tamoxifen to induce nuclear translocation of the $ER^{T2}$-Cre-$ER^{T2}$ and thus the recombination of the LoxP sites in the RITE system. Presence of Centriolin-NeonGreen was detected both inside and outside the ventricle, indicating successful recombination (*Figure 2B*). Interestingly, we observed centrosomes with varying NeonGreen-to-tdTomato ratios, indicating centrosomes of varied ages. Such mixed tdTomato and NeonGreen centrosomes were observed at longer time points (>20 days) after recombination indicating that centriolin was remarkably stable on the centrosome, labelled with CEP164.

## Ventricular NPCs retain the older centrosome

To assess whether there is an asymmetric inheritance of older centrosomes, the RITE-tagged centrosome signal was compared between NPCs in the VZ and progeny that had migrated away from the centre of cortical units. NPCs in VZ were selected for comparison, as opposed to all SOX2-positive cells, as their presence in the VZ strongly indicates their capacity for self-renewal whereas SOX2-positive cells outside of the VZ may have already started differentiation along the neural lineage.

Twenty-two days after recombination organoids were stained and the ventricles, and the surrounding tissue, were imaged (*Figure 2C*). During this process, the 3D tissue gets collapsed into a 2D image. As ventricle-like structures in organoids are tubular or spherical in nature, this dimension collapse produces a circle of centrosomes, where multiple Z planes show a ring (*Figure 2D* and *Figure 2—figure supplement 1B*). This leads to an inclusion of centrosomes whose-cell bodies are not imaged because of the radial orientation of NPCs relative to the ventricles and the volume of

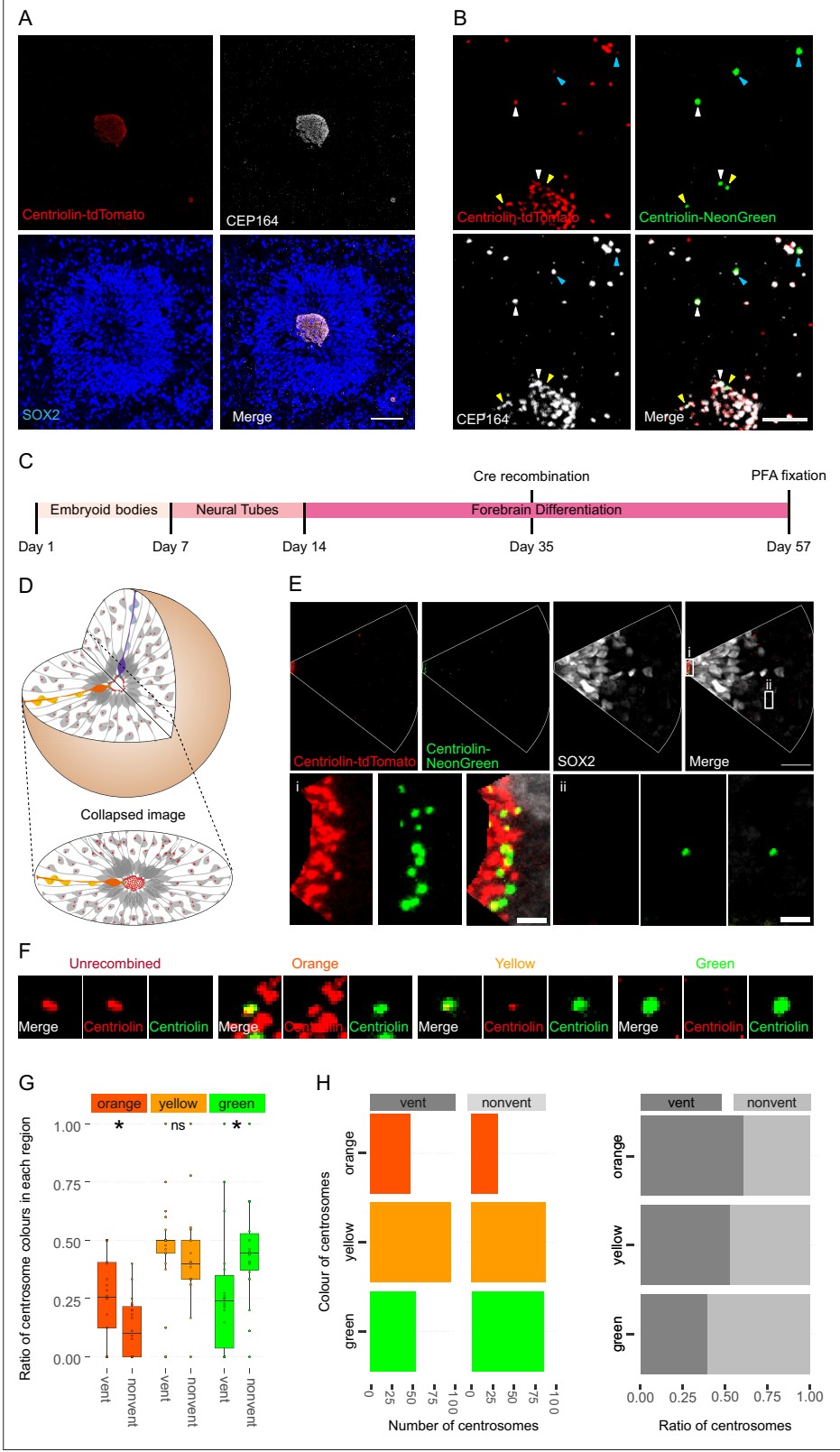

**Figure 2.** Asymmetric inheritance of centrosomes in human forebrain organoids. (**A**) Cortical unit of a day 35 forebrain organoid. The ventricular zone is easily identified by densely packed SOX2-positive neural progenitor cells (NPCs) that form a rosette-like structure around the ventricular centrosomes, shown by Centriolin-tdTomato and the mother centriole marker CEP164. Scale bar, 50 μm. (**B**) Recombined centrosomes exhibiting variable quantities

*Figure 2 continued on next page*

*Figure 2 continued*

of the older, Centriolin-tdTomato protein. Arrowheads indicate recombined centrosomes identifiable by the presence of Centriolin-NeonGreen. White arrowheads show centrosomes that have a large quantity of Centriolin-tdTomato present, thus are the oldest. Blue arrowhead indicated centrosomes have visibly less Centriolin-tdTomato, and the yellow arrowhead-depicted have no visible tdTomato signal, suggesting these are the most recently formed centrosomes. Scale bar, 5 µm. (**C**) Scheme showing the timing of Cre recombination in the context of the organoid protocol. (**D**) Schematic showing how imaging organoid sections causes the loss of progeny of some NPCs. In this example, two NPCs have recombined (orange and purple) and have produced progeny of a similar colour that migrate away along their radial processes. When imaging sections that include a ventricle, both the orange and purple NPCs' centrosomes are included in the image. The orange progeny are also included because they migrate away along the *x/y*-axis of the image. However, as the purple NPCs' progeny will migrate away along the *z*-axis, they are not imaged. (**E**) Example of an analysed image. The ventricular centrosomes are shown in large in (i) and an example of non-ventricular centrosome is shown in (ii). Scale bars, upper panel 20 µm, (i, ii) 2 µm. (**F**) Representative images that display the criteria of manual colour allocation. Recombined centrosomes signal was compared to unrecombined centrosomes. Centrosomes whose tdTomato signal was indistinguishable from unrecombined were allocated orange, those that had less tdTomato than unrecombined were yellow and those that had no tdTomato signal were green. Due to the presence of tdTomato signal, orange centrosomes would be the oldest, yellow the second oldest and green the youngest. (**G, H**) Analysis of the manual colour calling shows a significant enrichment of orange centrosomes in ventricle and a significant enrichment of green centrosomes outside of the ventricle (*n* = 22, cortical units). Cortical units defined as an entity within the organoid consisting of a ventricle-like structure surrounded by an inner ring of densly clustered NPCs and an outher ring of neurons. ns, non-significant, *p < 0.05.

The online version of this article includes the following source data and figure supplement(s) for figure 2:

**Source data 1.** Raw data related to *Figure 2G, H*.

**Source data 2.** Raw data related to *Figure 2—figure supplement 1C*.

**Source data 3.** Raw data related to *Figure 2—figure supplement 1D, E*.

**Figure supplement 1.** Centrosome analyses in organoids.

---

tissue imaged. This is problematic as the progeny of the recombined NPCs whose centrosomes are on the inner ventricular wall (shown in purple in *Figure 2D*) would migrate along the NPCs' radial projection and would not be in the imaged area. However, NPCs whose centrosomes are on the outer ventricular wall will produce progeny that should remain in the imaged area (shown in orange and yellow in *Figure 2D*). The inclusion of NPC centrosomes and not their progeny could potentially skew the data. To correct for this, images were digitally subdivided into areas of recombined NPCs and their likely progeny (*Figure 2E*). Centrosomes were manually allocated a colour by the quantity of Centriolin-tdTomato signal present in recombined centrosomes. Orange centrosomes' tdTomato signals were visibly indistinguishable from non-recombined centrosomes (*Figure 2F*); yellow centrosomes had some tdTomato signal but less compared to non-recombined centrosomes and green centrosomes showed no detectable tdTomato signal. The number of orange, yellow, and green centrosomes was counted in the VZ section (vent) and the non-VZ section (nonvent) and compared. Comparison between these two regions revealed a higher proportion of orange centrosomes in the ventricle compared to outside the ventricle (*Figure 2G, H*, *Figure 2—source data 1*). The inverse was observed for green centrosomes, whereas there was no difference seen in the localisation of yellow centrosomes, similar to what was observed in the absolute number of each centrosome colour (*Figure 2H*). To validate the manual approach, we used an unbiased method of analysing fluorescence levels at the centrosomes. The mean tdTomato and NeonGreen signal was acquired, and a ratio was calculated for each centrosome (see Methods for details). This analysis showed that VZ centrosomes had a lower proportion of NeonGreen signal compared to their non-ventricular progeny (*Figure 2—figure supplement 1C*, *Figure 2—source data 2*), corroborating the previous results obtained by manual grading. Next, centrosomes were divided into thirds by their NeonGreen to total signal ratio, with the highest, middle, and lowest thirds being labelled 'green', 'yellow', and 'orange', respectively. Again, we found increased orange centrosomes in the VZ and more green centrosomes in the non-ventricle areas within organoids (*Figure 2—figure supplement 1D–E*, *Figure 2—source data 3*). To test whether comparing ventricular to non-ventricular cells indeed reflects different cell types,

**Source data 2.** Raw data related to *Figure 3—figure supplement 1E, F*.

organoids were stained for NPCs (SOX2) and neuronal progeny (TBR1). TBR1 was excluded from the NPC dense VZ and indeed, green centrosomes co-localised with TBR1+ cells (*Figure 2—figure supplement 1F*). Taken together, these data indicate that the older, tdTomato-enriched centrosomes are preferentially retained by VZ NPCs, whereas the differentiating progeny inherits the younger, more NeonGreen-containing centrosomes.

## Ninein knockdown affects NPC fate

Next, we wanted to understand how centrosome inheritance affects the behaviour of human NPCs. Previous work identified that a knockdown of Ninein can randomise inheritance by preventing centriole maturation, without impeding the cells' ability to divide (*Wang et al., 2009*). We produced constructs that expressed human Ninein-targeting shRNA or a scrambled shRNA under the U6 promoter, as well as an H2B-CFP to facilitate identification of targeted cells (*Figure 3—figure supplement 1A*). To test the efficacy of the shRNA, we transfected HEK cells with the constructs and stained them for Ninein as well as centrosomal marker Pericentrin (*Figure 3—figure supplement 1C*). Analysis of the intensity of the Ninein signal showed a significant decrease between scrambled and Ninein-targeting shRNAs (*Figure 3—figure supplement 1D*, *Figure 3—source data 1*). Furthermore, HEK cells transfected with shRNA and GFP-Ninein (*Chen et al., 2003*) revealed a significant reduction in GFP-Ninein in cells expressing Ninein-targeting shRNA compared to scrambled shRNA (*Figure 3—figure supplement 1E, F*, *Figure 3—source data 2*). Ninein-targeting and control shRNA-expressing constructs were then electroporated into the VZ of day 35 WT human forebrain organoids (*Figure 3A*). After 5 days, we assessed the cell types of shRNA-targeted cells by co-staining with SOX2 or CTIP2 (*Figure 3B–D*, *Figure 3—source data 3*). Strikingly, within the H2B-CFP+ population (cells expressing either Ninein or scrambled shRNA) we found a decrease in the proportion of SOX2-positive cells upon Ninein knockdown, which corresponded with a significant increase in the neuronal, CTIP2-positive population. These data suggest that knockdown of Ninein, and therefore possibly randomisation of centrosome inheritance, leads to precocious neuronal differentiation of NPCs in the VZ (*Figure 3B–D*).

## Ninein knockdown alters centrosome segregation

As Ninein knockdown caused an increase in CTIP2-positive neurons, we next analyzed whether the older centrosome was still retained in the VZ or if it was also inherited by the differentiating daughter cell. To ensure constitutive expression of shRNA and recombination of only the cells expressing the shRNA, we generated constructs that expressed CRE-ER$^{T2}$ and the shRNA (*Figure 3—figure supplement 1B*) on a retroviral backbone. Day 24 organoids derived from Centriolin-RITE (CRE-ER$^{T2}$ negative) hESCs were transduced with either scrambled or Ninein-targeting shRNA retroviruses (*Figure 4A*). Organoids were induced with 4-OH tamoxifen, fixed 22 days later and their centrosomes colours were manually analysed (*Figure 4B*). Remarkably, we observed a shift in the localisation of the old centrosomes in the Ninein knockdown condition as compared to the scrambled, with an increase in old (orange) and a decrease in young (green) non-ventricular centrosomes in the Ninein knockdown cells as compared to the scrambled condition (*Figure 4C, D*, *Figure 4—source data 1*). This corresponded with the reverse seen in younger (green) centrosomes, where Ninein knockdown caused a substantial increase in green centrosomes in ventricular regions (*Figure 4C, D*). Taken together, these data suggest that Ninein plays a role in the inheritance of the older centrosome in the cells of the VZ and that aberration of Ninein, here through shRNA-mediated knockdown, leads to segregation of the older centrosome also into non-ventricular areas, associated with impaired NPC behaviour (*Figure 4E*).

## Discussion

We here identify accumulation of the older and more mature centrosome in the VZ, the residential area of NPCs in a model of the developing human cortex, which has hitherto not been described in human tissues. Such non-random and preferential inheritance of one of the centrosomes depending on centrosomal age by one of the two daughter cells has previously been reported for asymmetric cell divisions in yeast, *Drosophila*, and mice (*Wang et al., 2009*; *Januschke et al., 2011*; *Hotz et al., 2012*). The evolutionary conservation of this phenomenon emphasises the important role that centrosome inheritance may play in maintaining the proper cellular behaviour during asymmetric cell division.

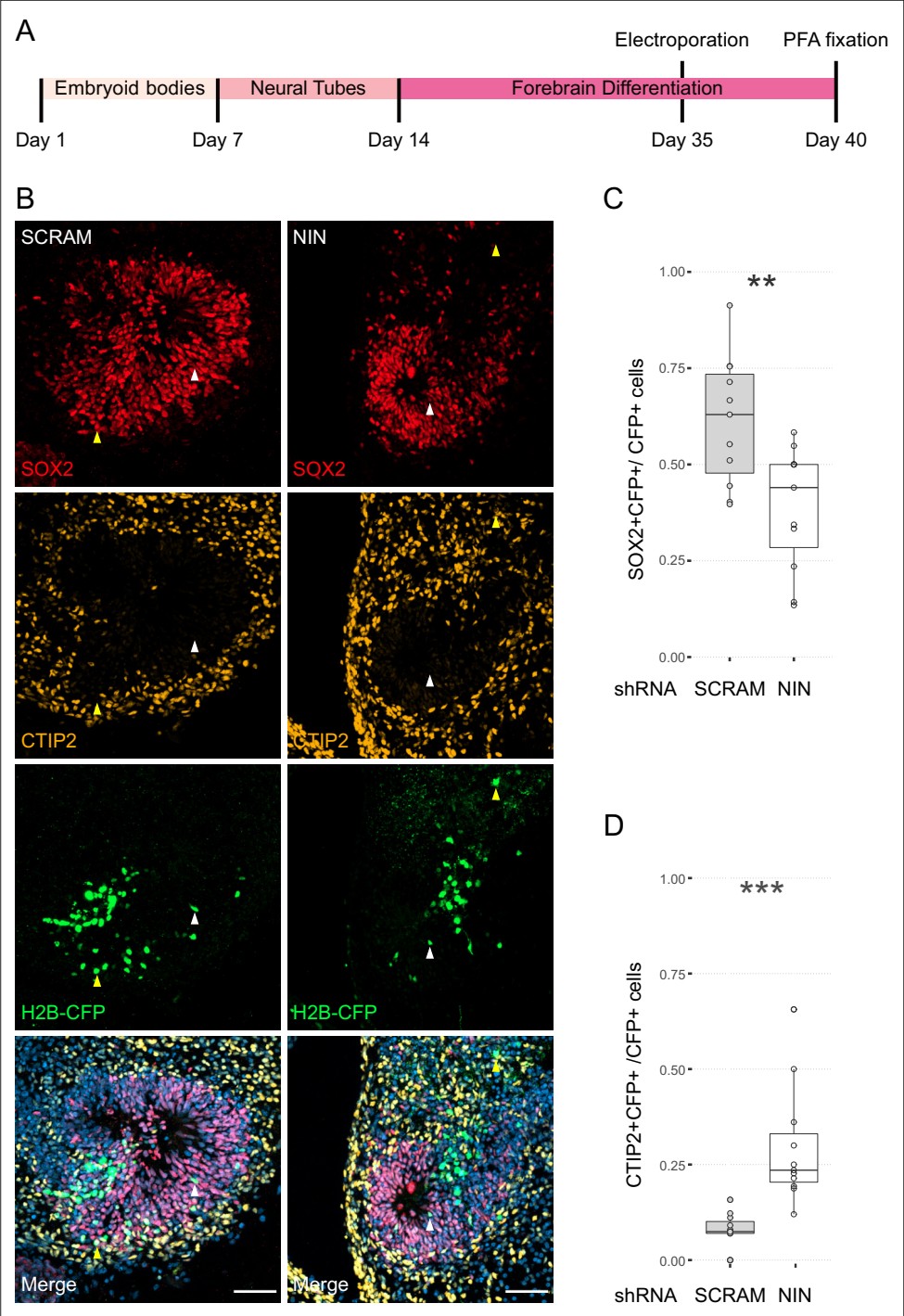

**Figure 3.** Randomisation of centrosome inheritance affects neural progenitor cell (NPC) fate. (**A**) Scheme showing the timing of electroporation of the shRNA construct in the context of the organoid protocol. (**B**) Day 40 WT organoids, 5 days post electroporation with either scrambled shRNA or NIN-targeting shRNA, and a H2B-CFP marker. Arrowheads depict H2B-CFP-positive cells that are either SOX2-positive (white) or CTIP2-positive (yellow). Scale bars, 50 μm. (**C**) Graph of the percentage of H2B-CFP-positive nuclei being SOX2-positive (*n* = 11, cortical units). (**D**) Graph of the percentage of H2B-CFP-positive nuclei being CTIP2-positive (*n* = 11, cortical units). **p < 0.01, ***p < 0.001.

The online version of this article includes the following source data and figure supplement(s) for figure 3:

**Source data 1.** Raw data related to *Figure 3—figure supplement 1D*.

*Figure 3 continued on next page*

*Figure 3 continued*

**Source data 3.** Raw data related to *Figure 3B–D*.

**Figure supplement 1.** Ninein knockdown in human cells.

In the study presented here, centriolin was endogenously tagged with fluorophores, in contrast to previous approaches relying on overexpression of centrosomal proteins. This reduces potential artefacts of excess protein expression; however, endogenous tagging provides only few fluorophores per centrosome to be visualised. Therefore, the signal is too dim to, with current technical approaches, perform live imaging or clonal analyses to follow individual centrosomes and their subsequent segregation. Nevertheless, the Centriolin-RITE tool kit presents a number of potential benefits over the previously described photoconversion-based approach (*Wang et al., 2009*). Photoconversion requires the tagged proteins to be targeted with a laser, which requires surgery or tissue extraction to allow for laser accessibility. This potentially alters the natural physiological processes within the tissues and may introduce experimental artefacts. Furthermore, photoconversion inevitably produces phototoxic events in cells which may alter their behaviour. The Centriolin-RITE birthdating tool can be activated in a spatial and temporal dependent manner by the use of tamoxifen-inducible Cre approaches. This allows any tissues within an organism to be birthdated without having to physically disturb it. Additionally, we designed the RITE system to use fluorescent proteins tdTomato and NeonGreen that originate from two evolutionary distinct species: *Discosoma* sp. and *Branchiostoma lanceolatum*, respectively (*Shaner et al., 2004*; *Shaner et al., 2013*). This facilitates biochemical separation of the birthdated proteins by immunoprecipitation; further, the addition of high-affinity biochemical tags to the RITE cassette is feasible and will allow for further biochemical and proteomics-based analyses, which are essential to untangling the complex protein–protein interactions in centrosomes (*O'Neill et al., 2022*). The data we present here show that asymmetric inheritance of centrosomes is an evolutionary conserved mechanism during cortical formation that may be of critical relevance for human brain development by maintaining the proper cellular behaviour of human NPCs. Furthermore, using it in human brain organoids will allow future experiments studying whether centrosomal age differs between apical and basal radial glia cells, the latter one being rare in brains of lisencephalic animals (*Camp et al., 2015*; *Pollen et al., 2015*).

The reason why human ventricular NPCs retain the older mother centrosome remains unclear. Several hypotheses may explain preferential inheritance of the old centrosome to the self-renewed daughter NPC. One potential reason could be that inheritance of the old centrosome may reduce aberrant mitotic events as the mother centrosome had participated previously in at least one successful mitosis, suggesting that the centrosome is capable of producing a daughter centriole, nucleating microtubules, and facilitating normal chromosome segregation, while the newly synthesised daughter centrosome is newer and less 'tested' as it has never produced a functional centrosome. If this centrosome has an aberration in its structure, it could lead to duplication defects and deleterious chromosomal missegregation. Thus, one may hypothesise that evolution would favour retaining the older, tested centrosome with the daughter cell that must undergo additional rounds of cell divisions. In support of this notion, the older centrosome is inherited by self-renewing NPCs in mouse (*Wang et al., 2009*) and, as we here show, in human cortical development, and by the daughter cell in budding yeast (*Pereira et al., 2001*; *Lengefeld et al., 2017*), which is considered the stem cell. In contrast, *Drosophila* neuroblasts inherit the younger centrosome and pass on the older centrosome to their differentiating daughter cells (*Januschke et al., 2011*). Why *Drosophila* neuroblasts do not follow the mode of centrosome inheritance observed in other species is not known. One may speculate that *Drosophila* neuroblasts produce less progeny and therefore have lower selective pressure to maintain their fitness.

Another potential reason for asymmetric inheritance could be that the old centrosome of NPCs is maintained on the ventricular wall associated with the primary cilium. Indeed, it has been shown that the mother centrosome is quicker to reassemble the primary cilium, likely because it was already decorated with the distal appendages from the previous interphase (*Paridaen et al., 2013*), which has been shown to be critical for NPC cellular behaviour. Moreover, at the ventricular wall, space is limited, and the apical feet of ventricular NPCs are crowded together forming adherens junctions. The stability of these junctions and the maintenance of the apical foot itself is dependent on the junctional microtubules (*Kasioulis et al., 2017*), which can be destabilised by increased centrosomal

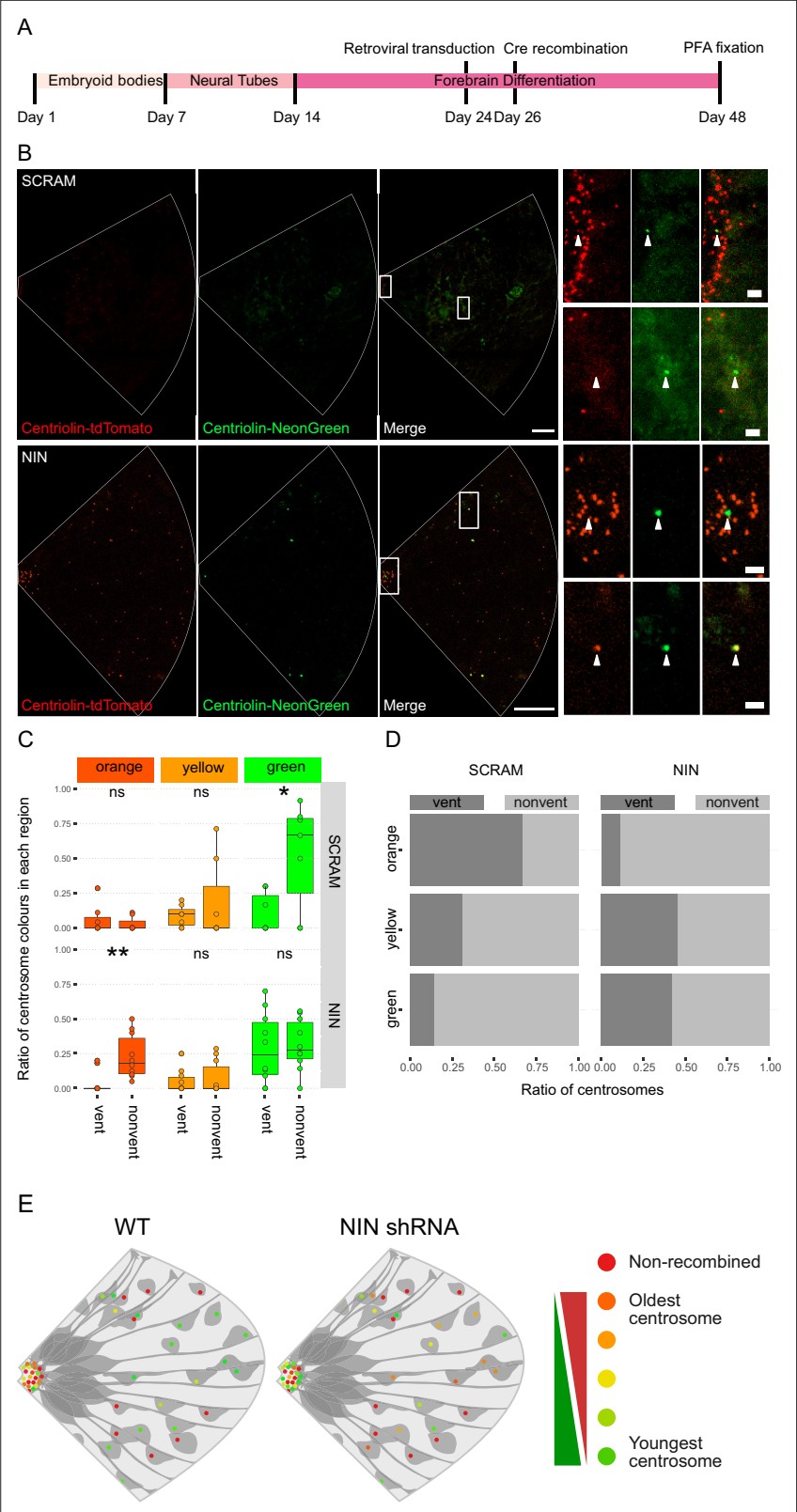

**Figure 4.** Ninein-shRNA alters retention of the older centrosome in the ventricular zone. (**A**) Timing of the retroviral transduction and subsequent recombination in the context of the organoid protocol. (**B**) Day 24 Centriolin-RITE organoids were infected with retroviral expression of scrambled or NIN-targeting shRNA, and Cre-ERT2. Organoids were fixed 22 days after recombination was induced with tamoxifen. Scale bars, 25 μm (left

*Figure 4 continued on next page*

*Figure 4 continued*

column), 2 µm (right column). (**C**) Comparison of the colour of centrosomes in each region by shRNA treatment using manual colour calling (scram n = 7 cortical units, nin = 10, cortical units; note that single unit data overlay for 'vent' data points). (**D**) Analysis of the colour composition of each region shows a shift in localisation of the older centrosomes to the nonvent region for the NIN shRNA-treated organoids (*n* = 8, cortical units). ns, non-significant, *p < 0.05, **p < 0.01. (**E**) Scheme of asymmetric centrosome segregation in WT and how it is perturbed upon Ninein knockdown.

The online version of this article includes the following source data for figure 4:

**Source data 1.** Raw data related to *Figure 4C, D*.

microtubule organisational activity (*Camargo Ortega et al., 2019*). Indeed, previous work showed that overexpression of the centrosomal protein AKNA increased centrosomal microtubule organisational activity, destabilised the apical junction, and caused NPCs to delaminate from the apical surface and migrate out of the VZ. Furthermore, the mother centrosome was shown to anchor to the apical membrane via the distal appendage protein CEP83 (*Shao et al., 2020*). Loss of CEP83 causes impaired apical anchorage of the centrosome, disorganisation of microtubules, increase in NPC proliferation and subsequent enlargement of the cortex. These data indicate that the maintenance of NPCs in the VZ requires a delicate balance of centrosomal function at the apical surface. Inheritance of the older centrosome could be one mechanism by which NPCs regulate this balance. Removing Ninein from the centrosome could alter this balance and lead to the older centrosome being inherited to a non-ventricle destined cell. Additionally, it is becoming apparent that centrosomes act as one of the key signalling centres of the cell (*Arquint et al., 2014*). The Notch signalling regulator, Mind-bomb1, was shown to be preferentially enriched on the daughter centrosome and asymmetrically segregated with it into the differentiating daughter cell (*Tozer et al., 2017*). Mind-bomb1 in this cell would then promote stemness via Notch signalling activation in the surrounding NPCs. Consistent retention of particular centrosome allows of such asymmetric signalling pathways to be established and could facilitate the coordination of other cellular asymmetries (*Royall and Jessberger, 2021*).

The extent to which proper asymmetric inheritance of the older centrosome by dividing NPCs is associated with human disease remains currently unknown. For example, disturbance of asymmetric inheritance of centrosomes could play a causative role in microcephaly where indeed a substantial number of human variants and mutations have been identified in genes associated with centrosome function and structure (*Faheem et al., 2015*). Strikingly, a recent cell type-specific analysis of centrosomal proteomes identified a large number of proteins, previously implicated with neurodevelopmental disease, to be associated with centrosomes (*O'Neill et al., 2022*). We here identify that human NPCs show asymmetric inheritance of centrosomes and that disrupting retention of the pattern of centrosome inheritance affects proper neurogenesis in human forebrain organoids. Together with these proof of principle findings, the genetic approach we present here may become a powerful tool in elucidating the role centrosome inheritance plays in human disease and cortical malformations.

## Methods
### Genetic targeting and constructs
Guide RNAs and CRISPR/Cas9 tagging was performed as described previously (*Denoth-Lippuner et al., 2021*). ENSEMBL (https://www.ensembl.org) was used to find the full sequences of the gene of interest and the terminal stop codon. The Zhang group guide design tool (https://www.crispr.mit.edu) was used to find best gRNAs, based on the closeness to the stop codon and the lowest number of off-targets. Guides were synthesised from Microsynth AG, Switzerland and cloned into pSpCas9(BB)-2A-Puro (Addgene 48139) following the cloning strategy described by *Ran et al., 2013*. The following gRNAs were used:

Human CNTRL: ACAAGACAGTATTCCTCATC
Human ER^T2^-Cre-ER^T2^: GGGGCCACTAGGGACAGGAT

To generate the homology arms required for the insertion of the construct via homologous-directed repair, an upstream and downstream region immediately adjacent to the terminal stop codon of the gene of interest was PCR amplified. Primers used were:

Human CNTRL
Upstream homology Forward: GCCTCTTTAATGTGCCCAAG
Upstream homology Reverse: TCTGGCTGAGGCATTCTTTTC
Downstream homology Forward: TGAGGAATACTGTCTTGTGTAAATATATTC
Downstream homology Reverse: CTTGGTGGTGAGGGATGACT

Upon establishment of a cell line, genomic DNA was extracted using QIAGEN DNeasy Blood & Tissue Kit (QIAGEN 69504). Primers to test correct integration of the construct were designed such a way that one primer was within the inserted, non-native DNA and one was outside the inserted DNA including the homology arms. Additional primers were designed that would span the whole inserted region; these would test for homozygous vs. heterozygous integration. PCRs were performed on WT and genetically modified genome DNA. CAG-Cre (Addgene No. 13776) was used for recombination via electroporation. AAVS1-T2A-Puro_CAG-ER$^{T2}$-Cre-ER$^{T2}$ was cloned by replacing DR-GFP from pAAVS1-DR-GFP (Addgene No. 113193) with ER$^{T2}$-Cre-ER$^{T2}$ fragment was obtained from CAG-ER$^{T2}$-Cre-ER$^{T2}$ (Addgene 13777) via restriction digest. shRNAs were taken from broad institute database (https://portals.broadinstitute.org/gpp/public/gene/search) and cloned into a retroviral backbone; viruses were produced as described before (*bin Imtiaz et al., 2021*). Later, H2B-CFP was removed from the constructs and replaced with CRE-ER$^{T2}$ that was taken from CAG-ER$^{T2}$-Cre-ER$^{T2}$ (Addgene 13777).

## Human ESCs and organoids

All hESC experiments were approved by the Kantonale Ethik-Kommission (KEK) of the canton of Zurich, Switzerland. H9 hESCs (WiCell) were maintained in feeder-free conditions and grown at 37°C with 5% $CO_2$ (*Thomson et al., 1998*). hESCs were fed with mTeSR1 or mTeSR plus (Stem Cell Technologies) in the absence of antibiotics and grown on hESC qualified Matrigel (Corning) coated plates. ReLeSR (Stem Cell Technologies) was used for routine passaging as it promotes stemness. Passaged cells were kept in media containing 10 µM Y-27632 (Stem Cell Technologies) for 24 hr to promote survival. Some protocols required a single-cell suspension of hESCs (e.g., electroporations, aggrewells); to obtain a single-cell suspension, hESCs were instead passaged with Accutase (Sigma-Aldrich). For freezing down, hESCs were resuspended in CryoStore CS10 (Sigma-Aldrich) and stored below −170°C in liquid nitrogen. For electroporations, hESCs were maintained in media containing Y-27632 for at least 1 hr to improve cell survival. Approximately 2 million cells were used per electroporation. Chilled Nucleofector V (Lonza) was used as the electroporation medium, and the electroporation was performed on the AMAXA electroporation system using the programme A-23. For overexpression, 1 µg of DNA per plasmid was electroporated. For gene editing, 4 µg of DNA per plasmid was electroporated. Recombination of Cre-ER$^{T2}$ expressing cells was induced in ER$^{T2}$-Cre-ER$^{T2}$ expressing cells by administration of 0.5 µM 4-hydroxytamoxifen (Sigma-Aldrich) for 24 hr. Neomycin resistant cells were selected with 100 µg/ml G418 sulfate (Gibco). Puromycin resistant cells were selected with 1 µg/ml puromycin (Gibco).

The human forebrain organoid protocol was described before (*Qian et al., 2016*; *Qian et al., 2018*). We used a modified form of this protocol, and the adjustments are as follows.

### Day 0, maintenance of hESC and embryoid body formation

hESCs were cultured in feeder free conditions. To produce EBs on day 0, a single-cell suspension was acquired with the use of Accutase. hESCs were incubated with 10 µM Y-27632 at least 1 hr prior to passaging to promote survival. The number of cells in the suspension was estimated with use of a cytometer and appropriate volume of suspension was added to AggreWell 800 (Stem Cell Technologies) that would result in 5000 cells per microwell being produced. AggreWells were pre-treated with Anti-adherence Rinsing Solution (Stem Cell Technologies) and AggreWell plates were centrifuged following the manufacturer's guidelines.

### Days 1–4, harvesting EBs and maintenance of EBs

On day 1, EBs are harvested following the AggreWell manufacturer's guide by gently pipetting with either a 5- or 1-ml pipette with the tip cut-off. Prior to harvesting, medium was ran the full length of the pipette to reduce the adherence of EBs to the side of the pipette. EBs were transferred to a

6 well Ultra-Low Attachment Plate (Corning) or a 10-cm Ultra-Low Attachment Plate (Corning) and maintained in mTeSR-E5 (Stem Cell Technologies) with 2 μM Dorsomorphin (Sigma-Aldrich) and 2 μM A83-01 (Tocris) until day 4. Media was changed days 3 and 4.

## Days 5–14, adaption and maintenance in induction media

Days 5–14 follow the exact same steps as described in *Qian et al., 2018*, but with a different media. The induction media used was mTeSR-E5 with 1 μM CHIR99021 (Stem Cell Technologies) and 1 μM SB-431542 (Stem Cell Technologies).

To ensure batch-to-batch variability was kept to a minimum, we started organoid production with the same number of cells each time and an organoid batch production was only started if the hESC culture looked healthy with very low levels of differentiation. In addition, only the H9 hESC line (purchased from WiCell) was used to remove genetic differences introducing batch effects. Furthermore, organoid batchs that did not show the signs of typical forebrain organoid development were not used for experimentation.

## Imaging and immunostaining

hESCs were fixed by either incubation with prewarmed 4% paraformaldehyde (Sigma-Aldrich) for 15 min at room temperature or with chilled methanol (Sigma-Aldrich) for 20 min at 4°C. All monolayer immunostainings used 3% Donkey Serum (Millipore) as a blocking agent and 0.25% Triton-X (Sigma-Aldrich) as a permeabilisation agent. Primary antibodies were incubated with the samples at 4°C overnight and secondary antibodies were incubated for 1 hr and 30 min at room temperature. Zeiss LSM 800 confocal microscope was used to obtain all images. For live cell imaging, hESCs were maintained at 37°C and with 5% $CO_2$ facilitated by the incubator chamber and heated stage that is fitted to the microscope. hESCs were imaged in Matrigel coated Lab Tek II chambered cover glasses (Thermo Fisher Scientific). For imaging following electroporation, cells were plated on the Lab Tek cover glasses. Imaging started after at least 6 hr. For imaging involving $ER^{T2}$-Cre-$ER^{T2}$ expressing cells, hESCs were first plated on Lab Tek chambered coverglasses (Thermo Fisher Scientific) and allowed to settle for at least 24 hr. Recombination was induced in situ with the addition of 0.5 μM 4-hydroxytamoxifen (Sigma-Aldrich) into the media. hESCs were imaged every hour. Throughout imaging hESCs were maintained in mTeSR plus containing 10 μM Y-27632 and penicillin–streptomycin–amphotericin B (Anti-Anti, Thermo Fisher Scientific). Additionally, media was supplemented with 1:1000 SiR-DNA (Spirochrome), in the absence of Verapamil, to visualise the nucleus and facilitate easier tracking of cells. Media was changed every 24 hr.

Organoids were electroporated as described (*Denoth-Lippuner et al., 2021*; *Denoth-Lippuner et al., 2022*). As each organoid was individually injected and electroporated, further technical repeats were not conducted. To determine the fate of electroporated NPCs expressing either scrambled shRNA or NIN-targeting shRNA, organoids were stained with antibodies detecting GFP (H2B-CFP), SOX2, and CTIP2. Every electroporated cell expressing shRNA (H2B-CFP) was categorised as NPC (SOX2+), neuron (CTIP2+), or undefined. For each cortical unit, the number of SOX2+ or CTIP2+ was divided by the number of targeted cells to assess the percentage of NPCs and neurons, respectively. The signal of H2B-CFP electroporated organoids was amplified using an anti-GFP antibody.

Recombination was induced in $ER^{T2}$-Cre-$ER^{T2}$ expressing organoids by administration of 10 μM 4-hydroxytamoxifen (Sigma-Aldrich) for 48 hr. Organoids were fixed with prewarmed 4% paraformaldehyde (Sigma-Aldrich) for 15 min at room temperature. Fixed organoids were suspended in 30% sucrose overnight at 4°C; organoids were maintained in 30% sucrose until sectioned. For sectioning, organoids were embedded in OCT compound (Tissue-Tek) and frozen solid. 20–40 μm sections were sectioned using a cryostat and adhered to cover slides. Sections were kept at −20°C until stained. For tissues containing fluorophores, proper measures were undertaken to prevent unnecessary and extensive exposure to light during the whole process. All organoid immunostainings used 10% Donkey Serum (Millipore) as a blocking agent and 0.5% Triton-X (Sigma-Aldrich) as a permeabilisation agent. Primary antibodies were incubated with the samples at 4°C for 1–3 nights and secondary antibodies were incubated for 1 hr and 30 min at room temperature. The signal of H2B-CFP electroporated organoids was amplified using an anti-GFP antibody.

## Image analysis

All images were analysed using ImageJ/Fiji and the data were processed and analysed in Excel or R. For organoid analysis, images with ventricular recombined centrosomes were selected. Cortical units were defined as a densely packed, spherical layer of SOX2+ cells, with their nuclei orientated inwards, surrounded by non-NSPC cells. To limit the effect of batch-to-batch variability, only well-formed cortical units were analysed. To define the size of the cortical unit, the thickness of the ventricle was measured using the SOX2 staining as the outer boundary. A circle with a radius of a maximum of three times the ventricle thickness or to the edge of the tissue (which ever was less), centred on the ventricle midpoint was cropped out. The centre of the ventricle was cropped out leaving a ring of three to five centrosomes, care was taken not to crop out recombined centrosomes that were part of the same cluster. A line was drawn between the two ends of the cluster. A perpendicular line was draw in the middle of the line extending radially out of the VZ to the edge of the circle. Two more lines were drawn from the ends of the first line at an angle of 45° extending to the edge of the circle. The area within these three lines and the arc of the edge of the circle was cropped out and analysed.

Recombined centrosomes were manually assigned one of three colours, orange, yellow, or green. The criteria for each were as follows and were dependent on the unrecombined centrosomes in the same image. Orange centrosomes were centrosomes with both tdTomato and NeonGreen signal, and where the tdTomato signal was similar to that of unrecombined centrosomes. Yellow centrosomes had both tdTomato and NeonGreen signal, but the tdTomato signal was visibly less than unrecombined centrosomes. Green centrosomes had only NeonGreen signal; the tdTomato signal in these centrosomes was undetectable by eye. The ventricular centrosomes (vent) and non-ventricular centrosomes (nonvent) were counted by colour for each image. Ratios of each colour were calculated and compared between different images. The absolute counts of centrosomes by colour were calculated across images and compared by their region. Additionally, the ratio of counts between regions was calculated for each colour.

Centrosomes were analysed in two groups: centrosomes from the ventricle (vent) and centrosomes outside the ventricle (nonvent). Vent centrosomes were defined as the centrosomes visibly localised on the ventricular wall, identifiable by the high density of centrosomes in a circular configuration. Nonvent centrosomes were centrosomes localised outside of the ventricular wall. Centrosomes were drawn around and the mean signal intensity was recorded for tdTomato and NeonGreen. The ratio of NeonGreen to total signal was calculated by dividing the NeonGreen by the sum of tdTomato and NeonGreen signal. Means of this ratio were calculate for each region of each image. This ratio was used for the digital allocation of orange, yellow, and green, with the centrosomes in the top third for highest ratio being assigned green, the second third yellow and the lowest ratio centrosomes being assigned orange.

## Validation of Ninein knockdown

HEK cells were transfected with constructs using Lipofectamine 2000 (Thermo Fisher Scientific) 2 days before they were fixed, stained, and imaged. Cells were either transfected with a plasmid expressing shRNA-targeting Ninein or scrambled shRNA. Cells were stained using antibodies against Ninein and Pericentrin and the ratio of their fluorescence intensity was measured in H2B-CFP-expressing cells. Alternatively, cells were additionally infected with a plasmid expressing GFP-Ninein (*Chen et al., 2003*) and stained for Pericentrin. Fluorescence intensity of GFP-Ninein and Pericentrin was measured in H2B-CFP-expressing cells and ratios were plotted.

## Used antibodies:

| Antibody | Species | Manufacturer | Catalogue No./RRID | Dilution |
|---|---|---|---|---|
| PCNT | Rabbit | Abcam | ab4448 RRID:AB_304461 | 1:1000 |
| SOX2 | Rabbit | Millipore | AB5603 RRID:AB_2286686 | 1:200 |
| SOX2 | Mouse | R&D | MAB2018 RRID:AB_358009 | 1:200 |
| CTIP2 | Rat | Abcam | ab18465 RRID:AB_2064130 | 1:200 |

*Continued on next page*

*Continued*

| Antibody | Species | Manufacturer | Catalogue No./RRID | Dilution |
|----------|---------|--------------|--------------------|----------|
| CEP164 | Rabbit | Abcam | ab221447/n.a. | 1:200 |
| tdTomato | Goat | Origene | AB8181-200/n.a. | 1:750 |
| Ninein | Mouse | Santa Cruz | sc-376420 RRID:AB_11151570 | 1:250 |
| GFP | Chicken | Aves | GFP.1020 RRID:AB_10000240 | 1:1000 |
| TBR1 | Rabbit | Abcam | ab31940 RRID:AB_2200219 | 1:250 |

## Statistical analysis

Statistical significance of all data presented here was tested by using unpaired, Student's *t*-test. Significance is represented with asterisks and ns, which correspond to the following p values: ns = p > 0.05, *p < 0.05, **p < 0.01, ***p < 0.001. The number of analysed data for the control and the test condition were kept the same. The number of analysed data represented in the figure legend as *n*, where *n* is equal to the number of control or test data points.

## Materials availability

All newly generated constructs and genetically modified hESCs are freely available after execution of appropriate materal transfer agreement. Requests should be directed to S.J. (jessberger@hifo.uzh.ch).

## Acknowledgements

We thank Y Barral for critical conceptual input and Y-R Hong for sharing the plasmid expressing GFP-Ninein. Funding Information: This work was supported by the European Research Council (STEMBAR to SJ), the Swiss National Science Foundation (BSCGI0_157859 and 310030_196869 to SJ), the Boehringer Ingelheim Fonds (to LNR), the URPP Adaptive Brain Circuits in Development and Learning (AdaBD) of the University of Zurich (UZH), and the Zurich Neuroscience Center. Funding sources were not involved in study design, data collection, and interpretation, or decision to submit the work for publication.

## Additional information

### Funding

| Funder | Grant reference number | Author |
|--------|------------------------|--------|
| European Research Council | STEMBAR | Sebastian Jessberger |
| Swiss National Science Foundation | BSCGI0_157859 | Sebastian Jessberger |
| Swiss National Science Foundation | 310030_196869 | Sebastian Jessberger |
| Boehringer Ingelheim Fonds | | Lars N Royall |

The funders had no role in study design, data collection, and interpretation, or the decision to submit the work for publication.

### Author contributions

Lars N Royall, Conceptualization, Formal analysis, Investigation, Visualization, Methodology, Writing - original draft, Writing - review and editing; Diana Machado, Formal analysis, Investigation; Sebastian Jessberger, Conceptualization, Funding acquisition, Writing - original draft, Writing - review and editing; Annina Denoth-Lippuner, Conceptualization, Visualization, Writing - review and editing

### Author ORCIDs
Lars N Royall http://orcid.org/0009-0004-5401-6726
Diana Machado http://orcid.org/0000-0003-0645-5084
Sebastian Jessberger http://orcid.org/0000-0002-0056-8275
Annina Denoth-Lippuner http://orcid.org/0000-0003-0357-5208

### Decision letter and Author response
Decision letter https://doi.org/10.7554/eLife.83157.sa1
Author response https://doi.org/10.7554/eLife.83157.sa2

## Additional files

### Supplementary files
• MDAR checklist

### Data availability
Data generated are included in the main and supporting files (Figure 2—figure supplement 1 and Figure 3—figure supplement 1, and the source data files containing the numerical data used to generate the figures).

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
