## [Editor Report]

The fundamental work that shows the preferential inheritance of the older centrosomes by the self-renewing daughter cells in human is supported by strong evidence. The findings will be of interest to developmental neurobiologists, but also more broadly to cell and developmental biologists.

---

## [Decision Letter]

**Decision letter after peer review:**

Thank you for submitting your article "Asymmetric inheritance of centrosomes maintains stem cell properties in human neural progenitor cells" for consideration by *eLife*. Your article has been reviewed by 3 peer reviewers, including Anita Bhattacharyya as the Reviewing Editor and Reviewer #1, and the evaluation has been overseen by Marianne Bronner as the Senior Editor.

The reviewers agree that the manuscript reports important findings in the field of developmental neurobiology, particularly in our understanding of human cortical development. The methods, data, and analyses support the claims with one key piece of data missing. Specifically, the lack of clonal resolution or timelapse imaging makes it hard to assess whether the inheritance of centrosomes occurs as the authors claim.

Essential revisions:

1) The authors should include analysis at clonal resolution or timelapse imaging to show the asymmetric inheritance of centrosomes during organoid development.

*Reviewer #1 (Recommendations for the authors):*

The authors state that differentiating progeny inherit younger, more neon green centrosomes, based on the localization of labelled centrosomes and distance from VZ (Figure 2). To directly support the conclusion that differentiating progeny inherit younger centrosomes, the authors need to show neuron double labeling to validate this point (either TF as in Figure 3, or neuron-specific marker).

The discussion needs to be reorganized and expanded.

1. The methodology should be discussed first and include improvements to existing technology.

2. Asymmetric inheritance generally (the bulk of the current Discussion).

3. Expanded discussion of human-specific aspects of NPCs to put the results in a better context. For example, the discussion should include aRG versus bRG and implications for the findings of the paper on human brain development.

*Reviewer #2 (Recommendations for the authors):*

1. To show the asymmetric inheritance of centrosomes during organoid development, the authors could perform live imaging on the organoid slices. The Reviewer acknowledges the effort authors did to quantify the centrosomes in their organoids but finds it important to corroborate such data with time-lapse imaging that can show beyond any doubt such asymmetric inheritance. Fully aware of the technical issues that might arise during such imaging, I do not find it necessary to repeat all the quantifications using live imaging, but I do think that showing movies with asymmetric inheritance would significantly increase the impact of this study.

*Reviewer #3 (Recommendations for the authors):*

1) In many images, red and magenta are used together. This makes the details hard to distinguish and especially for colour blind readers.

2) A small suggestion would be to interchange figure 3 with 4, the authors would first show that Ninein KD affects the segregation of new vs old centrosomes and then the consequence of the mis-segregation. It would improve the flow of the paper.

3) The explanation of the RITE-based tagging approach could be clearer, especially if Figure 1A could have a more detailed schematic of which cells would inherit the green and red dots.

---

## [Author Response]

Essential revisions:1) The authors should include analysis at clonal resolution or timelapse imaging to show the asymmetric inheritance of centrosomes during organoid development.

We very much understand the request for clonal or time-lapse imaging of RITE-labeled centrosomes in organoids. We do agree that such data would strengthen our findings. Indeed, we have tried to image individual RITE-labeled centrosomes in human organoids before. However, we have – despite very extensive efforts – failed to do so for the following reasons: in complex, highly scattering tissues, such as organoids, the RITE-based signal is too dim. This is due to the fact that the sizes of the fluorescent spots are extremely small (given the size of centrosomes just a few pixels consisting of very few centriolin molecules) which makes it obviously complicated and requires, for any even “dim” detection, very high power laser settings. However, the required laser power in turn causes extensive photobleaching (together with potential other effects on imaged tissues). Thus, we would have loved to add these data, have tried very hard before submitting the initial version of our manuscript, but we are afraid that these experiments are just not technically feasible at this time.

We understand that others have established very recently clonal linage tracing approaches in human organoids (e.g., He et al., 2022, Nature Methods). However, they used strong overexpression of GFP throughout the cell (which will be hundredfold brighter than the fluorescence signal of individual centrosomes). Indeed, we are not aware of any experiments showing long-term imaging of clonal lineages together with (very small) intracellular organelles, which will be needed to address the suggested revision experiment.

Purely for technical reasons, as outlined above, we will not be able to add novel data using RITE and clonal lineage tracing. We tried these experiments before but failed due to current technical limitations. Unfortunately, we had not discussed that shortcoming (and the technical difficulties) in our initial version. However, we now explicitly discuss in a revised version why i) such experiments would strengthen our data but also ii) why these experiments are not feasible at this time (page 13). Again, we fully agree that these data would strengthen our data; but at the same time we are convinced that the evidence provided by us is not just based on a “single observation” – but for example also very much supported by the Ninein-related experiments – and therefore are confident that the interpretation of our findings is valid based on the existing data and available technology.

Reviewer #1 (Recommendations for the authors):The authors state that differentiating progeny inherit younger, more neon green centrosomes, based on the localization of labelled centrosomes and distance from VZ (Figure 2). To directly support the conclusion that differentiating progeny inherit younger centrosomes, the authors need to show neuron double labeling to validate this point (either TF as in Figure 3, or neuron-specific marker).

We understand this point and have now included novel data showing that cells outside the VZ indeed express the neuronal marker TBR1 (Figure 2—figure supplement 1F). Please note the zoom-in images showing TBR1 expressing cells containing green Centriolin.

The discussion needs to be reorganized and expanded.1. The methodology should be discussed first and include improvements to existing technology.2. Asymmetric inheritance generally (the bulk of the current Discussion).3. Expanded discussion of human-specific aspects of NPCs to put the results in a better context. For example, the discussion should include aRG versus bRG and implications for the findings of the paper on human brain development.

We have modified our discussion accordingly. We changed the order and expanded the discussion of human-specific aspects. We did not discuss in more detail the impact of centrosome inheritance on the fate of aRG versus bRG as we used day 40 to day 60 organoids which might not contain substantial numbers of bRG yet. bRG were reported in day 80 organoids onwards (Qian et al., Cell, 2016).

Reviewer #2 (Recommendations for the authors):1. To show the asymmetric inheritance of centrosomes during organoid development, the authors could perform live imaging on the organoid slices. The Reviewer acknowledges the effort authors did to quantify the centrosomes in their organoids but finds it important to corroborate such data with time-lapse imaging that can show beyond any doubt such asymmetric inheritance. Fully aware of the technical issues that might arise during such imaging, I do not find it necessary to repeat all the quantifications using live imaging, but I do think that showing movies with asymmetric inheritance would significantly increase the impact of this study.

We do agree with the reviewer. And would have loved to add such imaging experiments. As outlined above (please refer to point #1 of the editorial summary), we have tried very hard to do these experiments. However, the size and fluorescent intensities of single centriolin-labeled centrosomes did not allow for the suggested experiments. We have explicitly discussed this shortcoming in our revised manuscript.

Reviewer #3 (Recommendations for the authors):1) In many images, red and magenta are used together. This makes the details hard to distinguish and especially for colour blind readers.

We have modified this.

2) A small suggestion would be to interchange figure 3 with 4, the authors would first show that Ninein KD affects the segregation of new vs old centrosomes and then the consequence of the mis-segregation. It would improve the flow of the paper.

This is an interesting suggestion, however, we felt like it is nice to end with the effect of Ninein knock-down on centrosome segregation, thereby, providing an answer to the original question.

3) The explanation of the RITE-based tagging approach could be clearer, especially if Figure 1A could have a more detailed schematic of which cells would inherit the green and red dots.

We now added a schematic to our revised manuscript summarizing the results in Figure 4E.